# Quadriceps Tendon Ruptures in Middle-Aged to Older Patients: A Retrospective Study on the Preoperative MRI Injury Patterns and Mid-Term Patient-Reported Outcome Measures

**DOI:** 10.3390/jpm13020364

**Published:** 2023-02-18

**Authors:** Kim Loose, Justus Rudolph, Martin Schlösser, Maximilian Willauschus, Johannes Rüther, Philipp Schuster, Hermann Josef Bail, Michael Millrose, Markus Geßlein

**Affiliations:** 1Department of Orthopedics and Traumatology, Paracelsus Medical University, Breslauer Straße 201, 90471 Nuremberg, Germany; 2Department of Radiology, Paracelsus Medical University, Breslauer Straße 201, 90471 Nuremberg, Germany; 3RKH Orthopedic Hospital Markgröningen, Center for Sports Orthopedics and Special Joint Surgery, 71706 Markgröningen, Germany; 4Department of Trauma Surgery and Sports Medicine, Garmisch-Partenkirchen Medical Centre, 82467 Garmisch-Partenkirchen, Germany

**Keywords:** knee surgery, tendon rupture, knee extensor mechanism injuries, magnetic resonance imaging

## Abstract

(1) Quadriceps tendon rupture (QTR) is a rare pathology, usually occurring in elderly patients with comorbidities, requiring surgical therapy. The aim of this study was to analyze rupture patterns and concomitant injuries using preoperative magnetic resonance imaging (MRI) *a*nd to evaluate patient-reported outcome measures. (2) In this retrospective cross-sectional study, 113 patients with QTR were screened and rupture patterns/concomitant injuries (*n* = 33) were analyzed via MRI. Clinical outcome was assessed in 45 patients using the International Knee Documentation (IKDC) and Lysholm score with a mean follow-up of 7.2 (±5.0) years. (3) The evaluation of preoperative MRIs showed multiple ruptures of subtendons in 67% with concomitant knee injuries in 45%. The most common associated pathology detected using MRI was pre-existing tendinosis (31.2%). Surgical refixation demonstrated good results with a mean post-operative IKDC score of 73.1 (±14.1) and mean Lysholm score of 84.2 (±16.1). Patient characteristics and individual radiologic rupture patterns did not significantly affect the clinical outcome of patients. (4) Acute QTRs are complex injuries with common involvement of multiple subtendons. MRI imaging can be useful for achieving an accurate diagnosis as pre-existing tendinosis as well as concomitant injuries are common, and might be useful for providing an individual surgical strategy and improving outcomes.

## 1. Introduction

Quadriceps tendon rupture (QTR) is an uncommon injury with an incidence of 1.37 per 100,000 [1]. The mechanism of a rupture can be direct or indirect. The most common cause is a simple fall, but it is also seen because of car accidents or sporting activities. Spontaneous or even bilateral ruptures can be found as well in selected cases. The rupture of a healthy quadriceps tendon is rare, but QTR may sometimes occur in younger patients below the age of 40 years due to forceful eccentric muscle contractions during strenuous activities.

Most ruptures occur in men between 50 and 60 years old with certain comorbidities affecting tendon collagen structure [2]. Tendon injury patterns can vary within a range from partial ruptures of quadriceps’ subtendons to complete ruptures of all tendons forming the quadriceps tendon. While the majority of partial ruptures can be treated nonoperatively, complete ruptures usually require surgical therapy to avoid severe functional deficits [3,4]. 

In addition to older patient age and male gender, known patients’ risk factors for QTR include rheumatoid arthritis, systemic lupus erythematosus, diabetes mellitus, long-term corticosteroid therapy, fluoroquinolones treatment, chronic kidney disease, second or tertiary hyperparathyroidism, gouty arthritis and peripheral vascular disease [5]. Pre-existing tendinosis is common in these patients, but its exact development is not fully understood, although it may play an important role regarding surgical refixation [6]. Fatty tendon degeneration and collagen alterations due to chronic inflammation are frequently described [7]. Furthermore, data on concomitant knee injuries are also scarce [2]. As QTR is often linked to falls due to sudden loss of motor control, concomitant injuries are likely to occur. Especially when flexion and rotational movements are combined during trauma, additional intraarticular injuries such as meniscal injuries or rupture of the anterior or posterior crucial ligament may occur. Resulting persistent instability and pain can cause inferior individual outcomes. Due to the demographic development, an increasing incidence in QTR can be expected. Therefore, the improvement of diagnostic accuracy as well as further investigation of different surgical techniques seems mandatory to advance an individual therapeutic approach. In a review by Arnold et al., it was noted that the additional information provided by magnetic resonance imaging (MRI) helps in visualizing the local anatomy and may be of benefit, although the incidence of concomitant injury remains relatively low [8].

The understanding of precise quadriceps tendon anatomy is crucial for successful surgery as surgeons should be aware of the trilaminar structure combining rectus femoris (RF) as a superficial layer, an intermediate layer formed by the vastus medialis (VM) and vastus lateralis (VL) muscles and a deep layer formed by the vastus intermedius (VI) muscle [9,10]. 

During surgery, the identification of the exact rupture pattern can be demanding and misinterpretation or insufficient debridement of tendinosis within the stump can lead to inadequate fixation resulting in a loss of function or re-rupture. Therefore, knowledge of rupture patterns and tendon quality seems essential for careful pre-surgical planning. Regarding exact rupture patterns in QTR, there have been few studies so far and, besides clinical examination, sonography is still the common method of choice for initial diagnosis [11]. However, the sensitivity and specificity of sonography in the assessment of quadriceps injuries can vary and it is known to be operator-dependent. In contrast, MRI is more standardized and can consistently and accurately visualize a ruptured tendon [8,12,13]. The recent literature has suggested that MRI is the gold-standard test for diagnosing QTR, with reported sensitivity, specificity, and a positive predictive value of 1.0 [8,12]. Falkowski et al. evaluated 53 QTRs via magnetic resonance imaging (MRI) in 2021 and found 183 subtendon ruptures affecting different parts of the quadriceps tendon (RF, VM, VL, VI) mainly localized near the patella [13]. Furthermore, MRI can be useful for describing tendon quality at the rupture site as well as bony spurs at the insertion and, therefore, for planning optimal surgical therapy [13]. A systematic review by Oliva et al. examined the literature on QTR and demonstrated results of chronic and complex QTR. They defined complex ruptures as re-ruptures (9%), chronic (16.5%) or linked to total knee arthroplasty (74.5%). They concluded that macroscopic tendon quality, rather than time since trauma, was critical in the choice of the surgical technique [14]. As MRI sequences have the capability to demonstrate individual tendon quality at the rupture site in each individual patient, it can be assumed that surgical therapy might benefit from this additional information.

Good clinical outcomes after QTR have been reported previously in several studies. Patients returned to their pre-injury activities despite advanced age and regardless of surgical technique used for fixation [15,16]. However, only few studies have focused on a middle-aged to older population [2,16]. As QTR represents a severe injury in the elderly resulting in a loss of mobility, quality of life and associated co-morbidity, further studies are needed in this special study population.

The primary aim of this study was to analyze rupture patterns and concomitant injuries in preoperative MRIs. The secondary aim of the study was to evaluate whether rupture patterns or patient-related characteristics have an impact on patient-reported outcome measures (PROMS) after surgical refixation. 

## 2. Materials and Methods

### 2.1. Study Population

From a total of 254 patients with acute QTR treated with surgical refixation in the period from January 2000 to August 2022 at a maximum care hospital, 113 patients were included in the study for a retrospective analysis (Figure 1).

Inclusion criteria were a minimum patient age of 40 years and surgical treatment of QTR. Exclusion criteria were concomitant injuries that affected the ipsi- or contralateral lower extremity, psychiatric disorders (including dementia), patients with complications, patients with existing knee implants or prior surgery on the knee, as well as re-rupture of the QT. Of the 113 patients eligible for the study, 15 were dead, 19 patients suffered from psychiatric disorders (including dementia) at the time of evaluation, and 4 patients had complications of treatment. Furthermore, 30 patients had to be excluded due to insufficient medical records.

### 2.2. Diagnosis and Surgical Treatment

Diagnosis in all patients was performed at the emergency department including history, clinical examination, plain X-rays, and ultrasound or MRI examination. The applied surgical techniques were transosseous tendon refixation using patellar drill-holes or refixation using suture anchors. In all cases, surgery was performed by an experienced orthopedic surgeon using a vertical incision starting at the proximal patella pole and extending it proximally. Tendon rupture patterns were explored and refixation was performed on all identified ruptured quadriceps’ subtendons after debridement of tendon stumps and creating a bony rim at the proximal patella pole. Postoperatively, the affected leg was put in an orthosis with the leg fully extended. All patients were mobilized with crutches or a walker on the second day after surgery under instructions from a competent physiotherapist. Patients were instructed to wear the knee straightening brace for 6 weeks. Physiotherapy was allowed after one week with passive ROM 0-0-30 and an increase in flexion for 30 degrees every two weeks. Full knee flexion and active knee extension were allowed after 6 weeks.

### 2.3. Evaluation of Preoperative MRI

Preoperative MRI was available for evaluation with 33 patients. Images were acquired using 1.5 or 3.0 Tesla scanners. The standard MR protocol consisted of sagittal proton density (PD) fat-saturated and sagittal PD non-fat-saturated images, coronal PD fat-saturated images, axial T2 fat-saturated images and axial T1 non-fat-saturated images.

Images were evaluated retrospectively by an experienced musculoskeletal consultant radiologist on an approved PACS workstation (Ashvins, MedicalCommunications, Heidelberg, Germany). For each quadriceps tendon rupture, the radiologist analyzed the injured subtendons of the quadriceps tendon as follows: RF, VM, VL and VI, as described in previous studies [13], and whether it was partial or complete. Furthermore, the exact location of rupture (myotendinous junction, intratendinous area and near the patella pole) was evaluated. In the case of complete rupture, the retraction distance was determined in millimeters (mm). Other concomitant injuries such as meniscal lesions, cruciate ligament ruptures and collateral ligament injuries, as well as signs of degenerative changes in the collagenous tendon structure such as tendinosis, were also evaluated.

### 2.4. Patient Related Data and PROMS

Patient-related data such as gender assigned at birth, age at surgery in years, length of hospital stay before and after surgery in days, American Society of Anesthesiologists Physical Status Classification System (ASA PS classification I-VI), body mass index (BMI) and prescribed medication were extracted from hospital medical records.

After hospital discharge, the clinical outcome was assessed by an experienced orthopedic surgeon in the outpatient clinic using the International Knee Documentation Committee (IKDC) and the Lysholm score [17,18], with a minimum follow-up of 1 year.

### 2.5. Statistical Analysis

All data were obtained and analyzed retrospectively. The statistical analysis was performed using Python (version 3.9.7. with SciPy 1.8.1, Python Software Foundation in Wilmington Delaware, USA). All *p*-values were statistically significant at *p*-value < 0.05. Normal distribution was tested using the Shapiro–Wilk test. For normally distributed data, the Pearson correlation coefficient was used. For non-normally distributed and normal-scaled data, the Spearman test was used. The Kruskal–Wallis H-test and Mann–Whitney U test were utilized depending on the group size. Unless otherwise stated, descriptive results are demonstrated as mean ± standard deviation and range. 

The study was approved by the institutional review committee (Paracelsus Medical University at the Nuremberg Hospital, Number IRB-2022-007) following national legal guidelines.

## 3. Results

Of 113 patients with acute, first-time QTR included in the study, 97 (85.8%) were male and 16 (14.2%) were female. Complete follow-up including IKDC and Lysholm scores was obtained in 45 patients (39.8%) with a mean follow-up of 7.2 (±5.0) years. 

### 3.1. Demographic Data and Patients’ Characteristics

The characteristics of the study population including ASA PS classification, number of regularly prescribed long-term medication, time-to-surgery and time spent in hospital are demonstrated in Table 1. 

### 3.2. Quadriceps Tendon Rupture Patterns

The majority of QTRs were localized near the patella pole (71.0%) at the bony junction between the quadriceps tendon and the patella (without bony avulsion). A small proportion of ruptures was found at the myotendinous junction (12.5%) and only few ruptures were considered to be intratendinous (9.30%). In two cases, ruptures could be found in two different localizations in the same patient. One rupture could not be precisely assigned using MRI. All identified rupture patterns are demonstrated in Figure 2.

Altogether, the 33 evaluated MRI images showed 70 ruptures of subtendons (RF, VM, VL and VI). While 62.9% were complete ruptures, 37.1% partial ruptures could be identified (Figure 3). Complete rupture in the rectus femoris (30.0%) was the most common (Figure 4a), followed by complete rupture of the vastus medialis (15.7%). When combining the affected layers of the quadriceps tendon with the location, complete distal rupture of the RF subtendon was most common. Most patients (67%) showed multiple ruptures of subtendons compared with single tendon ruptures (33%).

For complete QTR, the mean retraction distance was 16.5 mm (range 4.0–36.0 mm). The most common associated pathology with QTR detected via MRI was pre-existing tendinosis (31.2%). This was more frequent in complete ruptures (33.3%) than in partial ruptures (22.2%).

### 3.3. Concomitant Injuries Found in Patients with Quadriceps Tendon Ruptures

The evaluation of MRI images showed concomitant injuries with QTR in 15 patients (45.0%, Figure 5). Meniscal lesions were identified at the medial meniscus in one patient (3.1%) and at the lateral meniscus in four patients (12.5%). Partial collateral ligament lesions (34.4%) and partial lesions of the anterior and posterior cruciate ligaments (28.1%) were also detected.

### 3.4. Evaluation of PROMS after Refixation of Quadriceps Tendon Ruptures

As shown in Table 2, the overall patient-related outcome measured with the IKDC score (73.1 ± 14.1) and the Lysholm score (mean 84.2 ± 16.1) was good. The investigated patients’ characteristics did not show a significant correlation with clinical outcome (Table 3). Regarding the surgical technique, transosseous refixation (48%), tendon suture (27%) and the combination of transosseous refixation and intratendinous suture (25%) showed no influence on the measured outcome. Furthermore, the evaluation of rupture patterns also yielded no significant effect on PROMS after refixation of QTR with the described techniques. Re-ruptures of the QT after surgical therapy during follow-up occurred in 11 patients (9.7%). None of these patients had preoperative MRI. The percentage of patients who suffered from arterial hypertension as a cardiovascular disease was 72%, and 27% suffered from diabetes mellitus type II and obesity.

## 4. Discussion

The main finding of this study is that most patients showed a complex rupture pattern with the involvement of different subtendons via MRI. Concomitant knee injuries were common, as well as pre-existing tendinosis. Patient-related characteristics and individual radiologic rupture patterns did not significantly affect the clinical outcome of patients having surgical treatment for QTR. Re-ruptures were more frequently found in male patients without prior MRI diagnostics.

Most patients in this study showed multiple ruptures of the quadriceps´ subtendons compared to only single-tendon ruptures. When compared to previous studies, Falkowski et al. found 183 subtendon ruptures (RF, VM, VL and VI) via 52 MRIs when two radiologists independently evaluated the ruptures [13]. Our study is in concordance with the finding that the superficial quadriceps tendon layer (RF) is more frequently affected than the middle and deep layers [13]. Furthermore, complete ruptures were more often found than partial ruptures on the evaluated MRI images. One must keep in mind that only surgically treated QTRs were included in the study, while conservative therapy may have been used for partial ruptures.

The most common rupture location found was 1–2 cm above the insertion at the patella [19]. Most QTRs in this study occurred close to the patella, which is also in line with other studies describing 84.8–93.6% of ruptures as being close to the patella [13]. A possible reason for this common site of rupture was described by Petersen et al. in 1999, as there is an approximately 30 × 15 mm oval vessel-free area about 10 mm above the patella on the quadriceps tendon [19]. In this study, rupture location and retraction distance did not significantly affect the patient-related outcome. Interestingly, the re-rupture rate was found to be 11% in the study population compared to studies including younger patients with reported re-rupture rates of 2% [2]. Oliva et al. also described 9% re-ruptures in a systematic review on complex QTR [14]. This finding may emphasize the need to identify previously described risk factors which can be addressed during surgery such as osteophytes near the patella pole or pre-existing tendinosis. Therefore, proper rim preparation and debridement of the proximal patellar pole and tendon stumps before refixation may be crucial to assure proper blood supply and provide proper conditions for tendon healing. As recent research found that women (61%) are more likely to have tendinopathy compared with men (34%), surgeons should pay special attention to tendon debridement in older female patients [12]. The reported results in this study indicate that MRI diagnostics can be useful for better identifying those individual factors and providing the best possible surgical planning as pre-existing tendinosis was frequently found in the examined, completely ruptured subtendons. For this reason, MRI diagnosis in QTR was considered to be the gold-standard diagnostic examination in the recent literature as the rate of clinical misdiagnosis ranges from 10% to 50% when compared with other radiological procedures such as ultrasonography [8]. Missing or underestimating the degenerative tendon changes during surgery may contribute to re-rupture. Therefore, when tendinosis is present in MRI, an end to end suture alone cannot be recommended in primary repairs in these cases. Nevertheless, the role of tendinosis as risk factor for QTR is not fully understood in older patients. While tendinopathy usually occurs in younger patients due to chronic eccentric tendon overload, there is limited literature that focuses directly on quadriceps tendinopathy mainly caused by microvascular damage due to systemic or local co-morbidities and medication interfering with tendon remodeling [6,20]. Pope et al. reported medical comorbidities such as renal disease and diabetes mellitus as causes of the pathologic degeneration of tendons [21]. There is evidence that the hypovascularity of tendons has been associated with progressive degeneration and subsequent rupture [22]. However, on the other hand, histological examinations conducted on chronic tendinopathy samples found many large blood vessels which were described as an ‘angiofibroblastic’ response [23].

There is a paucity of data regarding concomitant injuries in QTR. In the study by McKinney et al., 10% of patients investigated with QTR had concomitant intraarticular knee injuries, especially in high-energy trauma [24]. The anterior cruciate ligament, the medial meniscus, and the lateral meniscus were most often involved [24]. The analysis of rupture patterns in this study showed that QTR can be complex and highly variable. Since a high load of energy is needed to rupture the QT, it seems possible that concomitant injuries may occur. As eccentric quadriceps contraction is a typical injury mechanism for QTR, the found partial ACL/PCL injuries in this study may be considered as acute concomitant injuries due to the hyperextension of the knee joint. Concerning the significant number of meniscal lesions in this study, it remains unclear as to whether these were pre-existing degenerative lesions or acute injuries. Nevertheless, it is interesting that meniscal lesions were only found in combination with complete ruptures of the QTR in this study. As concomitant injuries showed no significant effect on clinical outcome after QTR, further studies are needed to evaluate whether the additional treatment of concomitant injuries might improve long-term outcomes.

Regarding patient-related characteristics, no influence on the clinical outcome was found in this study. In accordance with this study, Garner et al. also described a mean age of 61 years for patients with QTR [25]. Obesity has also been reported in the literature as a risk factor for QTR [26]. Demonstrated patients’ characteristics in this study indicate worse, even though not statistically significant, clinical outcomes for those with obesity. Advanced age and obesity have already been associated in the literature with an increased risk of rupture due to fatty degenerative changes, loss of collagen and tendon calcification [8]. In these conditions, MRI can provide additional information about tendon quality and muscular atrophy [14]. This detailed information can be helpful for surgeons to choose the best possible strategies for tendon debridement and refixation in an older and overweight population. 

Similar to this study, Rao et al. found time-to-surgery did not significantly affect clinical outcome in QTR [27]. Their study included 38 patients and compared the clinical outcome when surgery was performed within 7 days or 14 days [27], as compared to studies demonstrating inferior results after 3 weeks [25]. According to Arnold et al., delayed surgical intervention may lead to regression and atrophy of the quadriceps muscle. Outcomes of surgical repair or reconstruction may not be predictable for patients with chronic QTR and substantial fatty muscle infiltration [8]. Therefore, in cases of chronic tears, MRI can be a valuable tool in the assessment of fatty muscular atrophy and tendon retraction. However, further studies are needed to assess the role of muscular atrophy found via MRI on outcomes in these patients.

In the study by Boudissa et al., the Lysholm score including 50 knees after QTR averaged 93.7 [28]. Rao et al. studied 38 patients under 40 years of age with QTR and came up with a mean Lysholm score of 85.4 [27]. The outcome results for the evaluated middle-aged to older population in this study were similar with a Lysholm score of 84.2 (±16.1). Strikingly, there is a good clinical outcome in several studies after surgical treatment of QTR, although it has been shown to be a complex injury occurring in middle-aged to older patients with considerable risk factors [2]. Concerning the long-term outcome, recent studies have shown that up to 97% of working-age patients were able to return to work without reassignment after QTR. The authors cite direct, accurate diagnosis and surgical therapy with intensive rehabilitation as reasons for the good clinical outcomes [28]. Rao et al. reported a 63% return-to-play rate in 19 athletes under 40 years of age who had competed to varying degrees [27]. 

There was no difference in IKDC or Lysholm score between the different surgical techniques in this study population. Similar results have been previously described by Plesser et al. in a smaller group of patients comparing transosseous refixation to suture anchors [15].

### Limitations

The study has several limitations. Firstly, it is a retrospective study with several important limitations in comparison to a prospective study setting. Investigators depend on the availability and accuracy of the medical records. Therefore, the study population is subject to selection bias because of the excluded cases due to insufficient medical records. The involvement of different healthcare professionals in patient care may also contribute to less accuracy and consistency in the measurement of risk factors and outcome(s) throughout the study than that achieved with a prospective cohort study.

Secondly, it was conducted in a single center without a control group. There was no independently, randomized image review which could have introduced a reporting bias. As the study period of 20 years was quite long, the images were acquired with two different scanners over time which might have contributed to a bias in image interpretation. The number of retrospectively obtained IKDC and Lysholm scores is small compared to the larger total sample, and no preoperative PROMS were available. However, it has to be considered that primary QTR is a rare injury and this study examined a relatively large number of patients compared with previous investigations. No strength or function tests were performed to evaluate an objective clinical outcome. Nevertheless, this study covers a long follow-up period with standardized MRI imaging as well as surgical procedures. 

## 5. Conclusions

Acute QTRs are complex injuries with common involvement of multiple subtendons. MRI imaging can be useful for achieving an accurate diagnosis since pre-existing tendinosis as well as concomitant injuries are common. Despite good clinical outcomes after refixation, these factors should be considered prior to surgery to prevent re-rupture. Preoperative MRI evaluation might also be useful in providing an individual surgical strategy and to improve outcomes.

## Figures and Tables

**Figure 1 jpm-13-00364-f001:**
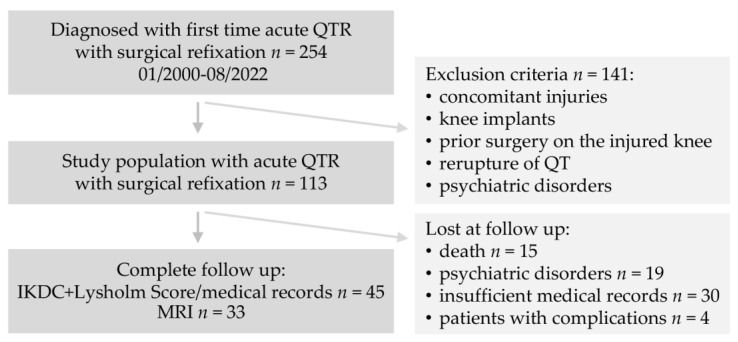
Study flowchart.

**Figure 2 jpm-13-00364-f002:**
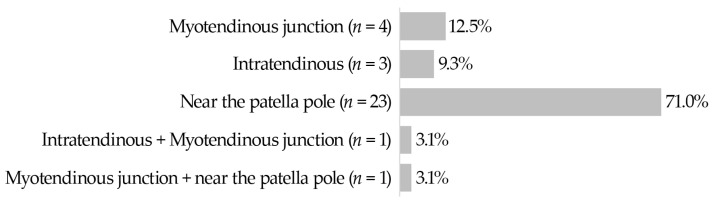
Rupture site based on MRI evaluation.

**Figure 3 jpm-13-00364-f003:**
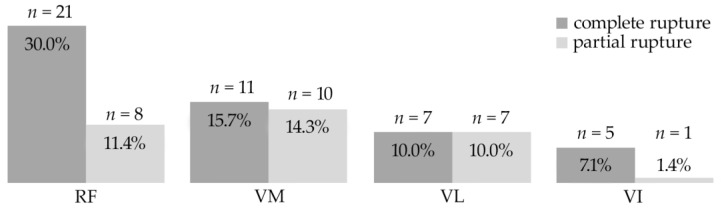
Partial and complete ruptures of quadriceps tendons by subtendons.

**Figure 4 jpm-13-00364-f004:**
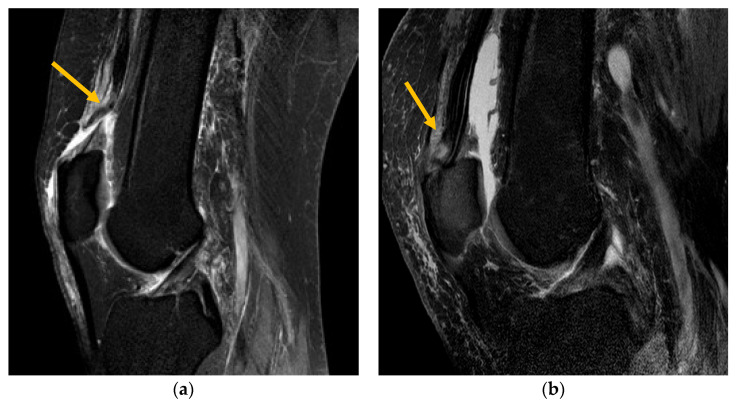
(**a**) T2-weighted sagittal MRI demonstrating complete QTR of the rectus femoris. (**b**) T2-weighted sagittal MRI demonstrating partial QTR of the rectus femoris subtendon.

**Figure 5 jpm-13-00364-f005:**
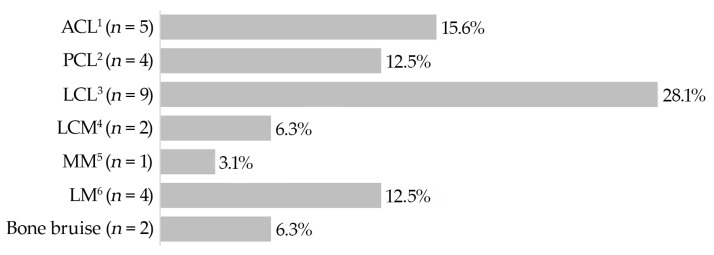
Concomitant injuries related to 33 patients via MRI after acute QTR. ^1^ Anterior cruciate ligament, ^2^ posterior cruciate ligament, ^3^ lateral collateral ligament, ^4^ medial collateral ligament, ^5^ medial meniscus and ^6^ lateral meniscus.

**Table 1 jpm-13-00364-t001:** Demographic data and patients’ related characteristics.

	All Patients (*n* = 113)	Female (*n* = 16)	Male (*n* = 97)
Age, y (mean ± SD)	61.2 ± 16.2	67.7 ± 17.0	60.1 ± 15.9
BMI ^1^ (mean ± SD)	32.5 ± 7.5	32.3 ± 7.7	32.5 ± 7.5
ASA classification ^2^ (*n*, %)			
ASA I	19 (16.8)	2 (12.5)	17 (17.5)
ASA II	57 (50.4)	7 (43.7)	50 (51.5)
ASA III	36 (31.8)	7 (43.7)	29 (29.8)
Time-to-surgery ^3^, d (mean ± SD)	7.6 ± 7.1	7.5 ± 7.8	7.6 ± 7.0
Hospital stay ^4^, d (mean ± SD)	4.9 ± 4.2	7.4 ± 4.9	4.5 ± 3.9
Number of medications, *n* (mean ± SD)	3.3 ± 2.9	4.1 ± 2.5	3.2 ± 3.0

^1^ BMI: body mass index; ^2^ ASA: American Society of Anesthesiologists Physical Status Classification System; ^3^ time: days from rupture to operation; ^4^ time: days from operation to hospital discharge.

**Table 2 jpm-13-00364-t002:** Follow-up and patient-reported outcome measure scores.

	All Patients (*n* = 45)	Female (*n* = 3)	Male (*n* = 42)
Follow-up in years (mean, range)	7.2, 1.0–17.1	8.4, 5.2–14.3	7.1, 1.0–17.1
IKDC score (mean, range)	73.1, 38.0–87.0	56.3, 38.0–84.0	74.3, 47.0–87.0
Lysholm score (mean, range)	84.2, 39.0–100.0	62.7, 39.0–99.0	85.9, 50.0–100.0
Re-ruptures (*n*)	11	-	11

**Table 3 jpm-13-00364-t003:** Correlation of patient characteristics and patient-related outcome.

Patients’ Characteristics for QTR	IKDC r-Value	IKDC *p*-Value	Lysholm r-Value	Lysholm *p*-Value
Age ^1^	−0.262	0.083	−0.274	0.069
BMI ^2^	−0.349	0.058	−0.304	0.102
ASA ^3^	1.467	0.480	1.339	0.512
Time-to-surgery ^1^	0.219	0.168	0.223	0.162
Time-after-surgery ^2^	−0.102	0.520	−0.026	0.872
Number of medications ^2^	−0.128	0.455	−0.122	0.477

^1^ Pearson correlation coefficient; ^2^ Spearman correlation coefficient; ^3^ Kruskal–Wallis H-test.

## Data Availability

Not applicable.

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
