# Peer review of "Quadriceps Tendon Ruptures in Middle-Aged to Older Patients: A Retrospective Study on the Preoperative MRI Injury Patterns and Mid-Term Patient-Reported Outcome Measures"

_jpm, 2023, doi:10.3390/jpm13020364_

Round 1
Reviewer 1 Report
1. Is there statistical significant correlation rupture location and fixation (example, where the fixation is more effective)? line 211
2. It is clear the distinction between partial and complete rupture in MRI?
(if not, can be MRI an accurate preoperative tool)
Reviewer 2 Report
The authors present a retrospective study investigating preoperative MRI QTR patterns/concomitant injuries and PROMS after surgical intervention.This is an area that has received a little attention in the literature, therefore, warrants further examination. Overall, the manuscript is written and organized fairly well. It follows the logical sequence of a research purpose. Despite this strength, I have few comments that need to be addressed by the authors and listed below.
TITLE. I would recommend to change the title by : “Quadriceps tendon ruptures in middle aged to older patients: A retrospective study on the preoperative MRI injury patterns and mid-term Patient-Reported Outcome Measures.”
ABSTRACT.
Authors should specify that 113 patients were screening but only 33 patients were used to determine the injury patterns with MRI and that 45 patients have been involved for the PROM.
INTRODUCTION.
The introduction is concise and clearly present the aim of the study. However, some key papers directly related with the aims of the study are missing and should be integrated in the introduction (and key points added when appropriated):
1) Oliva F, Marsilio E, Migliorini F, Maffulli N. Complex ruptures of the quadriceps tendon: a systematic review of surgical procedures and outcomes. J Orthop Surg Res. 2021 Sep 4;16(1):547. doi: 10.1186/s13018-021-02696-9.
2) Elattar O, McBeth Z, Curry EJ, Parisien RL, Galvin JW, Li X. Management of Chronic Quadriceps Tendon Rupture: A Critical Analysis Review. JBJS Rev. 2021 May 6;9(5). doi: 10.2106/JBJS.RVW.20.00096.
3) Arnold EP, Sedgewick JA, Wortman RJ, Stamm MA, Mulcahey MK. Acute Quadriceps Tendon Rupture: Presentation, Diagnosis, and Management. JBJS Rev. 2022 Feb 7;10(2). doi: 10.2106/JBJS.RVW.21.00171.
METHODS
No major points observed.
RESULTS
Results should only be presented for the patients with preoperative MRI available (n=33) and for patients with PROM available (n=45). The presentation of the 113 patients is not relevant related to the aim of the study and accessible MRIs and PROM data.
Figure 4 represents a preoperative MRI showing complete QTR in the rectus femoris subtendon. Authors should add a figure 5 illustrating, at the opposite, a partial QTR. In the MRI view, authors should insert an arrow locating exactly the observed lesion.
DISCUSSION
Major conclusions of the literature including missing studies (see specific comment for section Introduction) should be better linked with the current results. Authors should really enlightened their findings obtained with this retrospective study using MRI to revealed injury patterns in QTR.
4,1 Limitations. “Firstly, it is a retrospective study with the inherent limitations”. Please explain why.
